∂ | **Open Peer Review** | Bacteriology | Research Article

# Performance of the Xpert Xpress Strep A for detection of *Streptococcus pyogenes* from pediatric pleural fluids

Geneviève Amaral,[1] Vijay J. Gadkar,[1,2] Abdulrahman Almodahka,[1] David M. Goldfarb,[1,2,3] Ryan Penny,[2] Nicole Watson,[2] Suk Dhaliwal,[2] Tammy Chen,[2] Jocelyn A. Srigley,[1,2] Jonathan B. Gubbay,[1,2,3] Miguel Imperial,[1,2] Lynne Li[1,2]

**ABSTRACT** In Canada, Group A streptococcal (GAS) pneumonia is a severe invasive GAS (iGAS) condition reportable to public health for consideration of chemoprophylaxis for close contacts. It is often complicated by parapneumonic effusion. Sterile site culture from pleural fluid is the diagnostic gold standard but is insensitive, particularly given that many patients will have received antibiotics prior to sample collection. Xpert Xpress Strep A (Cepheid, Sunnyvale, CA) is a quantitative real-time (qPCR) assay targeting the *speB* gene designed to rapidly detect *Streptococcus pyogenes* from throat swabs. We evaluated its performance compared to bacterial culture for pediatric pleural fluids. A convenience sample of 36 specimens (21 positive, 15 negative) obtained January 2023 to April 2025 was included. Results were compared to a composite reference standard consisting of positive culture and a laboratory-developed (LD) qPCR targeting the *spy1258* gene. Limit of detection (LoD) for LD qPCR was determined by testing a dilution series of negative pleural fluids spiked with a predefined concentration (copies/mL) of *S. pyogenes* ATCC 19615 and using 95% hit rate analysis for Xpert. All 21 positive samples tested (100%) were detected by Xpert and the composite reference standard, while only 8/21 (38%) were culture positive. Xpert demonstrated 100% positive percent agreement, negative percent agreement, and overall categorical agreement (κ = 1) with the composite reference standard and individual LD qPCR. There was 64% categorical agreement between Xpert and culture alone (κ = 0.3). Xpert was more sensitive for detecting *S. pyogenes* than culture from pleural fluid. Faster turnaround time shortened the window to optimal therapy, supporting its diagnostic utility.

**IMPORTANCE** Group A streptococcal (GAS) pneumonia, a predominant pathogen in pediatric complicated pneumonia and a severe form of invasive GAS infection, requires prompt diagnosis, both for clinical patient management and identification of close contacts for public health evaluation. Pleural effusion is a common complication. Recovery of GAS from a sterile site like pleural fluid is a key criterion to categorize disease as invasive. However, microbiological yield is often low and may be improved by applying molecular diagnostic techniques. In our study, we demonstrate the utility of Xpert Xpress Strep A for Cepheid GeneXpert for rapid detection of GAS in pediatric pleural fluids. We demonstrated excellent sensitivity with Xpert detecting all 21/21 (100%) positive specimens, compared to culture alone, which was only positive for 8/21 (38%).

**KEYWORDS** GeneXpert, Xpert, GAS, *Streptococcus pyogenes*, pleural fluid, parapneumonic, effusion, empyema, pediatric

S*treptococcus pyogenes* is a common colonizer of human cutaneous and mucosal sites; however, it can also cause a spectrum of disease presentations ranging from mild (e.g., pharyngitis, cellulitis) to severe, including invasive Group A streptococcal (iGAS)

**Peer Reviewer** Tara D. Fuller, Indiana University School of Medicine, Indianapolis, Indiana, USA

Address correspondence to Lynne Li, lynne.li@cw.bc.ca.

D.M.G. received a speaker honorarium from Roche Diagnostics. J.B.G. is a paid consultant scientific editor for GIDEON Informatics, Inc. (https://www.gideononline.com/), which is unrelated to the current work. The authors declare no other conflicts of interest to disclose.

infection. Since becoming a nationally notifiable disease in Canada in 2000, the incidence of iGAS infections has been increasing, with infants, young children, and elderly adults most at risk (1–3). During the COVID-19 pandemic, incidence fell from a peak of 8.6 cases per 100,000 population in 2018 to 5.56 cases per 100,000 population in 2021 (2). In 2022, easing public health restrictions was associated with resurging iGAS cases in Canada (6.5 cases per 100,000, a 16.6% rise compared to 2021), as well as several other countries (2, 4). Along with *Streptococcus pneumoniae* and *Staphylococcus aureus*, *S. pyogenes* is a common cause of community-acquired bacterial pneumonia in the pediatric population and often progresses rapidly with associated large pleural effusions (1, 5). Increased rates of empyema attributable to iGAS were reported in several jurisdictions in 2022 when compared to pre-pandemic years (6–8).

Diagnostic confirmation is important both clinically for treatment optimization and carries public health implications in terms of close contact identification, post-exposure prophylaxis, and epidemiological surveillance (9). In Canada, chemoprophylaxis is recommended for close contacts of severe iGAS cases, defined as streptococcal toxic shock syndrome, soft tissue necrosis (necrotizing fasciitis, myositis, and gangrene), meningitis, GAS pneumonia, or death attributable to GAS infection (9). Importantly, chemoprophylaxis is time-sensitive and is most optimally administered to eligible contacts within 24 h (1, 9).

Microbiologic diagnosis from pleural fluid is important to facilitate optimal antimicrobial treatment (10). Although sterile site culture is the gold standard for diagnosis, prior studies indicate that, in general, bacterial culture yield from pleural fluid is quite low (15%) when compared with a composite reference standard of Gram stain, culture, *S. pyogenes* rapid antigen testing, and *S. pyogenes* PCR due to pre-collection antimicrobial exposure, limited pathogen load, or issues with transport or storage (5). A previous report showed that for pediatric patients with empyema, only 32% of those who are clinically diagnosed ultimately had a microbiological diagnosis made even after combining results from a variety of specimens (blood, pleural fluid, and lung tissue) (11). Interpretation of upper respiratory tract cultures can also be challenging to identify a causative organism, as organism recovery from these sites may reflect asymptomatic colonization (12).

Traditional culture-based identification methods generally require 18–48 h of incubation before identification is possible (13). Molecular assays provide an alternative diagnostic modality with increased sensitivity compared to culture (14). Among the main gene targets for identification of *S. pyogenes* are *spy1258*, which is a putative transcriptional regulator specific for *S. pyogenes*, and *speB*, a chromosomally encoded structural gene encoding the virulence factor streptococcal pyrogenic exotoxin type B (15–17). Laboratory-developed tests can be designed using these targets; however, this requires expertise and can be labor- and cost-intensive, with extraction and processing steps requiring several hours of run time.

Therefore, simple, rapid commercial tests requiring minimal hands-on time are a useful alternative to improve diagnostic pathways to support time-sensitive antimicrobial optimization for GAS pneumonia cases and close contacts. The Xpert Xpress Strep A for GeneXpert is a commercial quantitative real-time (qPCR) assay requiring minimal technical expertise. Xpert targets *speB*, designed to rapidly detect GAS from throat swabs with a manufacturer-reported sensitivity of 100% and a specificity of 94% for qualitative reporting (18). However, to our knowledge, its application for invasive specimens such as pleural fluid has yet to be reported. We compared its performance for detection of GAS from pediatric pleural fluids to a composite reference standard of culture and in-house laboratory-developed (LD) qPCR.

## MATERIALS AND METHODS

### Study setting and sample types

The study was performed at a tertiary pediatric academic center in Vancouver (BC, Canada). Both retrospective and prospective convenience samples of clinical pleural fluid were obtained from patients presenting to hospital between January 2023 and April 2025. One additional sample was included by spiking *S. pyogenes* ATCC 1915 into pleural fluid. Samples were stored at −80°C prior to testing.

### Test performance

The Xpert Xpress Strep A (Cepheid, Sunnyvale, CA) assay was performed using the GeneXpert instrument (Cepheid). Pleural fluid (300 µL) was applied directly to the single-use test cartridge and run on GeneXpert as per manufacturer's instructions ($n$ = 36) (18). The cartridge contains reagents for GAS detection, a sample processing control, and probe check control; the system integrates extraction, amplification, and detection via qPCR, and finally, a computer with preloaded software performs qualitative interpretation.

All Xpert results were compared to a composite reference standard (CRS), which included results from corresponding culture for each specimen and an LD qPCR targeting the *spy1258* gene based on a previously published method (19). We chose to include a nucleic acid amplification test targeting a separate GAS-specific gene target, *spy1258,* in the CRS, given the known poor sensitivity of culture alone for pleural fluid samples in the context of empyema (5, 11). A true positive result was therefore considered either the *spy1258* LD qPCR positive and/or culture positive, and a true negative was defined as both of these tests being negative.

The LD qPCR is performed using 5 µL of extracted pleural fluid sample combined with 20 µL Master Mix comprised of TaqMan Fast Advanced Master Mix (Thermo Fisher Scientific, Waltham, Massachusetts) and a primer (300 nM each)/probe (200 nM) mix using previously published CDC-GAS primer and probe sequences (20). Temperature cycling consists of: 95°C × 2 min, followed by a sequence of 95°C × 3 s, then 60°C × 20 s, repeated 45 times.

At our center, pleural fluid culture setup includes inoculation onto 5% sheep blood agar plate (BAP) and chocolate agar (CHOC) [incubation 35°C × 48 h in $CO_2$], MacConkey agar (MAC) [35°C × 24 h in ambient air], Brucella anaerobic agar (BAN) [35°C × 5 days in anaerobic conditions], and, if sufficient volume, a BD BACTEC Peds Plus blood culture bottle [35°C × 5 days] to maximize sensitivity. Isolates are identified by Matrix-Assisted Laser Desorption-Ionization Time-of-Flight (MALDI-TOF MS) (Bruker, Billerica, Massachusetts, United States).

Total nucleic acid was extracted from 350 µL of pleural fluid using QIAsymphony (Qiagen, Hilden, Germany) extraction platform. LD qPCR was performed using the QuantStudio Pro 6 instrument (Thermo Fisher Scientific, Waltham, Massachusetts, United States).

### Reproducibility and stability

Two strains, *S. pyogenes* ATCC 1915 (*emm*80) [spiked in negative pleural fluid at a concentration of 1:10 from stock 0.5 McFarland] and a clinical positive pleural fluid (*emm*12) were tested in single replicates on Xpert on days 0, 3, and 7. The spiked pleural fluid was kept at 4°C during the 1-week reproducibility testing period.

### Analytical sensitivity

A 0.5 McFarland standard was prepared from *S. pyogenes* ATCC 19615 colonies from BAP in 1 mL Liquid Amies (COPAN Diagnostics, Murrieta, California, United States), with extraction and LD-qPCR steps performed as above. The cycle threshold (Ct) value was mapped to a calibration curve of known concentrations (copies/mL [c/mL]) to obtain a

proxy defined starting concentration, from which a 2-fold dilution series was prepared using negative clinical pleural fluid as diluent. The calibration curve (Ct vs. $Log_{10}$ [c/mL]) to determine the concentration (c/mL) of the gDNA extracted from the whole bacterium was generated using a double-stranded, synthetic DNA fragment (gBlock, IDT, Coralville, IA) containing the *spy1258* target. This calibration curve provided the actual copies of the target (*spy1258*) in the gDNA extracted from the whole bacterium.

For the LD -qPCR, six dilutions were performed with a minimum of eight replicates tested for each dilution. The concentrations of each dilution beginning from the highest concentration were 103.34 copies/mL (c/mL), 51.61 c/mL, 25.83 c/mL, 12.9 c/mL, 6.45 c/mL, and 3.22 c/mL. The LoD was calculated using the Probit analysis model using MedCalc statistical software version 23.3.1 (Ostend, Belgium).

For the Xpert platform, six dilutions as outlined above with five replicates ($n = 5$) per dilution were performed (number of replicates was limited due to expense and a limited amount of available pleural fluid diluent) from the same prepared solution as above. LoD for the Xpert platform was defined as the lowest amount of the target that could be detected with a 95% hit rate analysis (21, 22).

## Analytical specificity (exclusivity) and reactivity (inclusivity)

Ten non-*S. pyogenes* organisms, including other streptococcal species and other pathogens implicated in empyema, were tested. Two clinical samples were used, comprised of a pleural fluid positive for *S. pneumoniae* and a polymicrobial sample with *Streptococcus anginosus*, *Enterococcus faecalis*, and *Klebsiella pneumoniae*. The remaining six organisms were clinical isolates spiked in 1 mL Liquid Amies media from BAP: *S. anginosus*, *Streptococcus intermedius*, *Streptococcus constellatus*, *Streptococcus dysgalactiae* (Lancefield type A), *Streptococcus mitis*, and *Arcanobacterium haemolyticum*. A 0.5 McFarland standard was prepared from BAP colonies and diluted with phosphate-buffered saline and spiked to an approximate concentration of $10^5$ CFU/mL in 1 mL Liquid Amies, which was then run on GeneXpert.

Six different known *S. pyogenes emm* type lineages were tested, including *emm*80 (*S. pyogenes* ATCC 19615 spiked in pleural fluid), *emm*1 (non-M1u$_K$ and M1$_{UK}$) and *emm*12 (both clinical pleural fluids), and *emm*2 and *emm*4 (clinical *S. pyogenes* culture isolates spiked in 1mL Liquid Amies).

## Statistical analysis

Positive, negative, and overall percent agreement (PPA, NPA, and OPA, respectively) were calculated using a 2 × 2 table (Table 1). Agreement was compared with the kappa statistic (κ). Average cycle threshold (Ct) values (and standard deviation) for clinical pleural fluid specimens were calculated using Microsoft Excel. A paired two-tailed *t*-test was performed to compare Ct values across molecular assays using an online statistical calculator (https://www.socscistatistics.com/) (23).

## RESULTS

### Comparison of methods

Twenty-one of 36 (58%) of specimens were positive for *S. pyogenes* by the composite reference method. Of these, culture was only positive in 8/21 specimens (38%), including the *S. pyogenes* ATCC 1915. In 2/20 (10%) of the LD qPCR-positive clinical specimens,

**TABLE 1** Performance of the Xpert Xpress Strep A in comparison to composite reference standard of LD qPCR *spy1258* and culture

| Xpert xpress strep A | Composite reference standard | | Total |
| --- | --- | --- | --- |
| | **Positive** | **Negative** | |
| Strep A detected | 21 | 0 | 21 |
| Strep A not detected | 0 | 15 | 15 |
| Total | 21 | 15 | 36 |

**TABLE 2** Performance of the Xpert Xpress Strep A in comparison to culture alone

| Xpert Xpress Strep A | Culture | | Total |
|---|---|---|---|
| | Positive | Negative | |
| Strep A detected | 8 | 13 | 21 |
| Strep A not detected | 0 | 15 | 15 |
| Total | 8 | 28 | 36 |

gram-positive cocci in chains had been observed in Gram stain but failed to grow in culture. The remainder of the LD-qPCR-positive, culture-negative specimens and the LD-qPCR-negative specimens demonstrated no visible organisms in Gram stain, with one exception showing polymicrobial flora attributable to non-GAS organisms. Xpert and LD qPCR detected *S. pyogenes* in all 21 known positive specimens; thus, Xpert demonstrated 100% PPA, NPA, and OPA (κ = 1) with CRS (Table 1). Compared to culture alone, Xpert demonstrated 100% PPA, 54% NPA, and 64% OPA (κ = 0.3) (Table 2). Statistics are summarized in Table 3.

## Reproducibility and stability

Xpert demonstrated 100% reproducibility and stability for detection of GAS across 7 days.

## Analytical performance

The 95% hit rate analysis for Xpert LoD was 25.8 copies/mL (c/mL), while the LD qPCR targeting *spy1258* LoD was $5.2 \times 10^2$ c/mL (95% CI: $3.4 \times 10^2$ – $8.9 \times 10^2$ c/mL). The average Ct value for positive clinical specimens (*n* = 20) was 26 (SD: 7.9) for Xpert, and 27 (SD: 5.4) for *spy1258* LD qPCR (Fig. 1). No statistically significant difference between the means was detected by paired *t*-test (*P* = 0.15).

Xpert demonstrated 100% inclusivity for all *emm* types tested and 100% analytical specificity with no cross-reactivity for any of the non-*S. pyogenes* organisms tested.

## DISCUSSION

Accurate diagnosis of iGAS pneumonia is essential to optimize antimicrobial treatment for patients, support contact tracing to facilitate chemoprophylaxis decisions, and support public health surveillance (1, 9). When complicated by parapneumonic effusion or empyema, pleural fluid can be sampled to make a microbiological diagnosis; however, traditional culture is limited by poor sensitivity when compared to molecular diagnostic methods (5, 11–14). Lack of a microbiological diagnosis could lead to unnecessarily broad antimicrobial therapy or additional invasive diagnostic sampling. Thus, molecular testing to improve diagnostic yield and treatment optimization is an attractive solution (12–14, 24).

To our knowledge, ours is the first published study utilizing Xpert Xpress Strep A for detection of GAS from pleural fluids. Xpert demonstrated 100% PPA, NPA, and OPA with the composite reference standard, consistent with its excellent performance for application to throat swabs (18, 25). All molecular methods were more sensitive than culture alone, in keeping with prior studies. A 2014 study in Ontario found that among 56 children admitted to hospital for CAP with parapneumonic effusion, 16% were caused by *S. pyogenes* (second to *S. pneumoniae,* accounting for 62%), and overall, molecular

**TABLE 3** Summary of Xpert Xpress Strep A performance statistics and test characteristics as compared to composite reference standard (CRS, LD -qPCR, and culture) vs. culture alone[a]

| | NPA (%) | OPA (%) [κ] | Accuracy (%) | Sn (%) | Sp (%) |
|---|---|---|---|---|---|
| Xpert vs. CRS | 100 | 100 (1) | 100 | 100 | 100 |
| Xpert vs. culture | 54 | 64 [0.3] | N/A | N/A | N/A |

[a]PPA, positive percent agreement; NPA, negative percent agreement; OPA, overall percent agreement; κ, kappa statistic; Sn, sensitivity; Sp, specificity; N/A, not applicable.

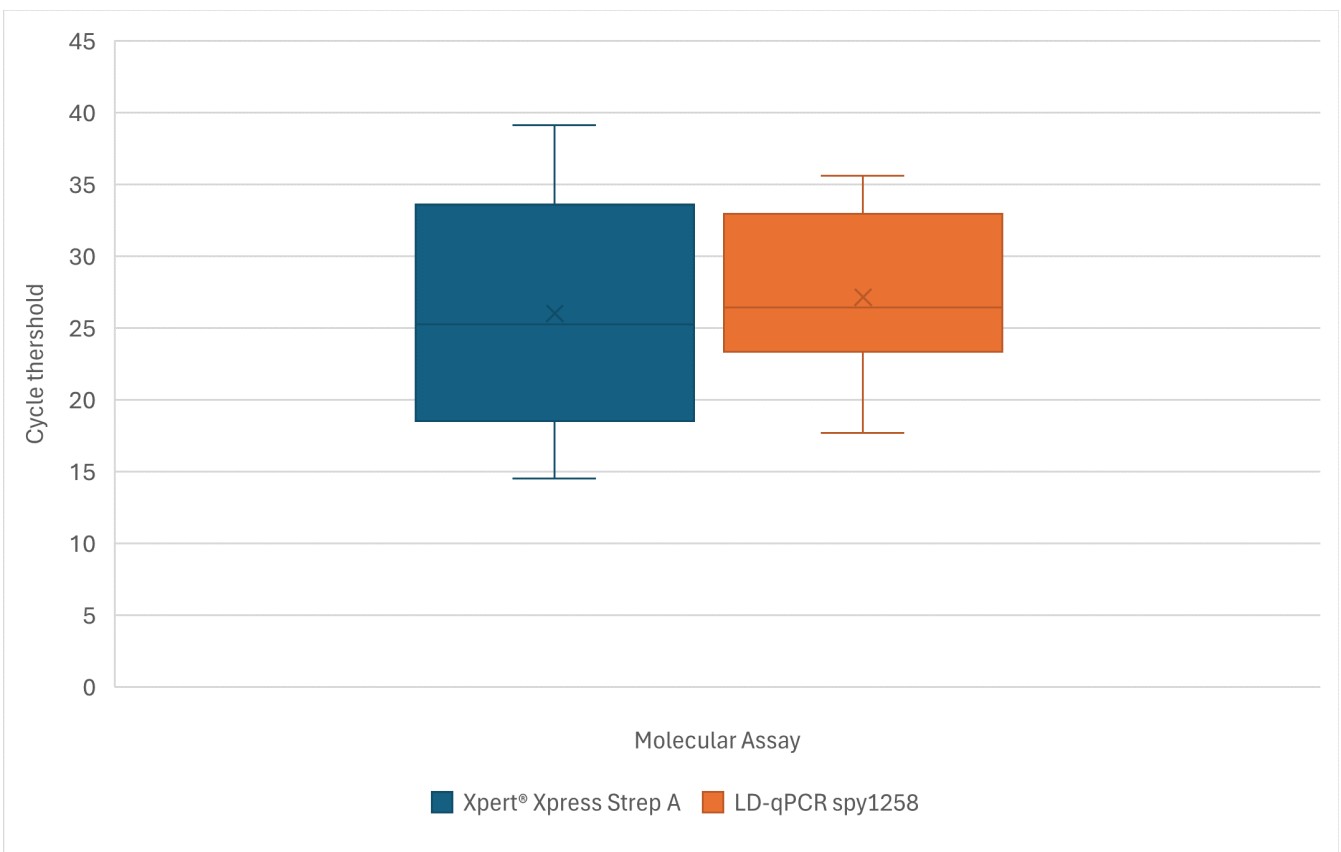

**FIG 1** Boxplot of cycle threshold value ranges between Xpert and LD qPCR *spy1258* for positive clinical pleural fluids (*n* = 20). No statistically significant difference between assays was observed by paired *t*-test (*P* = 0.15).

methods detected a pathogen in 82% of cases, compared to 25% by traditional culture (24). A further Ontario study in 2023 identified *S. pyogenes* as a pathogen in 14/109 (14%) of cases over a 5-year period, again second to *S. pneumoniae* (40/109; 37%), with most microbiologic diagnoses achieved through molecular testing of pleural fluids (as opposed to blood culture); importantly, definitive therapy with a narrow-spectrum agent was significantly associated with pathogen identification (10).

Xpert also demonstrated 100% inclusivity for six different *emm* types, including *emm*1 and *emm*12, which were the most predominant subtypes among the pediatric population <15 years of age in Canada in 2022 (25.8% and 24.2%, respectively) (2). Furthermore, the *emm*1 sublineage M1$_{UK}$ was detected, of importance in the setting of increasing global spread and toxigenicity of this M-protein at the time of our evaluation (26). Although the chromosomal *speB* gene is highly conserved among *S. pyogenes* and expected to be present across *emm* types, there have been prior reports of *emm*82 and *emm*11 isolates that were negative for *speB* by PCR (27, 28). Thus, this highlights the possibility of false-negative results; however, the rarity of these mutants suggests that assays targeting *speB,* like Xpert, will still detect the majority of GAS isolates.

Although all molecular methods improved sensitivity compared to culture alone and had comparable 95% hit rate and LoD, Xpert was more time-efficient, with results available in <25 min, including hands-on and automated-PCR run time, in keeping with prior studies (25). In our laboratory, in addition to sample transport time, LD qPCR consists of sample DNA extraction, PCR run time, and result reporting, which would take on average 24 h turnaround time due to daily sample batching. Xpert was the more cost-efficient molecular method in our setting compared with GAS LD qPCR due to low volume of testing, higher manual hands-on time, additional controls with each run, and external proficiency testing. In our laboratory, considering each of these additional

elements, the cost of LD qPCR versus Xpert would be $220 versus $44 CAD, respectively. To optimize our diagnostic approach, we have implemented Xpert for primary testing of pleural fluids for GAS concurrent with the traditional culture-based approach. As the faster molecular platform, if GAS is not detected on Xpert, specimens are reflexed to LD qPCR targeting *S. pyogenes* via a different target (*spy1258*) and *S. pneumoniae*, respectively. This multi-step molecular approach reduces the cost of diagnosing iGAS pneumonia but still enables targeted detection of two of the most common microbial etiologies of parapneumonic effusion in the pediatric population (1, 29, 30), including GAS isolates that are *speB*-deficient.

The widespread adoption of the GeneXpert instrument for other microbial diagnostic applications poises this technology for effective and easy implementation of adapting this method for GAS pneumonia diagnosis to other centers, particularly in areas where molecular qPCR methods are limited (31). While alternative rapid molecular methods such as loop-mediated isothermal amplification have been developed, a readily available assay for widespread implementation in the field has yet to be implemented (15). We did not evaluate the performance of Xpert compared to other available commercial multiplex PCRs, which offer respiratory tract panels with additional targets; however, the utility of other assays may be restricted by their inclusion or exclusion of the predominant pathogens implicated in complicated pediatric pneumonia/empyema, which are *S. pneumoniae*, *S. pyogenes*, and *S. aureus* (1, 32). For example, the BIOFIRE FILMARRAY Pneumonia Panel (bioMérieux, Marcy-l'Étoile, France) has 33 targets, including these three organisms among the 18 bacterial, 8 viral, and 7 antimicrobial resistance gene targets (32). However, it is not currently approved in our jurisdiction by Health Canada, and furthermore, other organisms besides the three main etiologies on this panel are less common causes of empyema in pediatric patients, which may not justify increased cost of additional targets (31). The BIOFIRE SPOTFIRE Respiratory/Sore Throat Panel has received Health Canada Approval and does include *S. pyogenes* as a target (33); however, currently, this assay costs more than Xpert and does not have *S. aureus* or *S. pneumoniae* as targets for our purposes. Thus, at this time, Xpert may be a more cost-effective, parsimonious method particularly for smaller or resource-limited hospitals.

However, there are limitations to the use of Xpert. First, low sample volumes may preclude the use of Xpert, which consumes a minimum of 300 µL with no residual extract to use for subsequent testing, potentially making test prioritization a challenge, especially in a pediatric population. By comparison, the LD-qPCR methods require 350 µL of initial neat sample to yield sufficient nucleic acid extract to be applied across multiple assays, which require 5 µL of the extracted volume for each assay. Furthermore, molecular detection is based on the presence of an applicable gene target and relies on accurate primers; an organism harboring a mutated *speB* gene/primer sequence could lead to false-negative results. This potential issue has been demonstrated with a different PCR assay for an isolate with a deletion in the *ropB* region (transcriptional regulator of *speB*) resulting in no amplification of *speB* (located in the same region as *ropB*) (34). Finally, Xpert provides only binary detection and does not predict antimicrobial susceptibilities, nor does it specify *emm* type. In our setting, molecular typing is conducted by the National Microbiology Laboratory (Winnipeg, Manitoba) and is performed reliably from clinical isolates only.

Our study was limited in scope in that samples were derived from a single clinical laboratory. Test performance on the small convenience sample of pleural fluids from pediatric patients may not be reflective of the general population. Furthermore, the LoD for LD-qPCR was determined using extrapolation from calculated starting concentrations, as opposed to true known concentration, possibly affecting accuracy of the LoD estimate, although all molecular methods evaluated still demonstrated superior clinical and analytical sensitivity compared with culture.

Future work should explore test performance on other sterile sample types, such as joint fluid or tissue biopsies. Furthermore, the clinical and public health outcomes related to using Xpert for iGAS diagnosis should be evaluated, and the utility of Xpert should

be compared to other commercial assays including multiplex PCRs. Finally, developing direct-from-specimen molecular typing as well as identifying molecular predictors of resistance for culture-negative PCR-positive cases would enable improved surveillance of the subtypes and resistance profiles of circulating severe iGAS strains.

## Conclusion

The Xpert Xpress Strep A provided accurate, rapid detection of GAS from pleural fluid, and its use should be considered to expedite clinical diagnosis of GAS pneumonia. PCR-based detection methods are more sensitive than culture for detection of GAS from pleural fluids, and Xpert Xpress Strep A was more time- and cost-effective than LD qPCR.

## ACKNOWLEDGMENTS

We wish to thank the technologists at BC Children's Hospital for their contributions to specimen procurement and processing.

A preliminary version of this study has been presented as a poster abstract at the AMMI Canada/CACMID annual conference in April 2025 (Calgary, AB, Canada). Cepheid was not involved in the study design, on-site experimentation, or analysis.

Study conceptualization & methodology (G.A., V.J.G., A.A., D.M.G., R.P., S.D., N.W., T.C., J.A.S., J.B.G., M.I., and L.L.); data curation and investigation (G.A. and V.J.G.); formal analysis (G.A., V.J.G., and L.L.); project administration (R.P., S.D., N.W., T.C., D.M.G., and L.L.); supervision (D.M.G. and L.L.); writing – original draft (G.A.); writing – review and editing (G.A., V.J.G., A.A., D.M.G., R.P., S.D., N.W., T.C., J.A.S., J.B.G., M.I., and L.L.).

All authors consent to the publication of this manuscript.

## AUTHOR AFFILIATIONS

[1]Department of Pathology and Laboratory Medicine, Faculty of Medicine, University of British Columbia, Vancouver, British Columbia, Canada
[2]Division of Medical Microbiology, Virology and Infection Control, Department of Pathology and Laboratory Medicine, BC Children's and Women's Hospital and Health Centre, Vancouver, British Columbia, Canada
[3]Division of Infectious Diseases, Department of Pediatrics, BC Children's Hospital, Vancouver, British Columbia, Canada

## AUTHOR ORCIDs

Geneviève Amaral http://orcid.org/0009-0003-5697-2909
Vijay J. Gadkar http://orcid.org/0000-0002-2676-3802
David M. Goldfarb http://orcid.org/0000-0003-0835-9504
Jocelyn A. Srigley http://orcid.org/0000-0002-0030-7665
Jonathan B. Gubbay http://orcid.org/0000-0003-0026-3786
Miguel Imperial http://orcid.org/0000-0002-4061-7940
Lynne Li http://orcid.org/0000-0002-1945-1173

## AUTHOR CONTRIBUTIONS

Geneviève Amaral, Conceptualization, Data curation, Formal analysis, Investigation, Methodology, Writing – original draft | Vijay J. Gadkar, Conceptualization, Data curation, Formal analysis, Investigation, Methodology, Writing – review and editing | Abdulrahman Almodahka, Conceptualization | David M. Goldfarb, Conceptualization, Methodology, Project administration, Supervision, Writing – review and editing | Ryan Penny, Conceptualization, Methodology, Project administration | Nicole Watson, Conceptualization, Methodology, Project administration, Writing – review and editing | Suk Dhaliwal, Conceptualization, Methodology, Project administration, Writing – review and editing | Tammy Chen, Conceptualization, Project administration, Writing – review and editing | Jocelyn A. Srigley, Conceptualization, Methodology, Writing – review and editing |

Jonathan B. Gubbay, Conceptualization, Methodology, Writing – review and editing | Miguel Imperial, Conceptualization, Methodology, Writing – review and editing | Lynne Li, Conceptualization, Formal analysis, Methodology, Project administration, Supervision, Writing – review and editing

## DATA AVAILABILITY

No data was submitted to a repository.

## ETHICS APPROVAL

Based on completion of the institutional project sorting tool based on a previously established ethics screening instrument, this was deemed a quality improvement project and did not require institutional research ethics approval (35).

## ADDITIONAL FILES

The following material is available online.

### Open Peer Review

**PEER REVIEW HISTORY (review-history.pdf).** An accounting of the reviewer comments and feedback.

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
