## [Reviewer comments · Microbiology Spectrum]

Microbiology Spectrum

Performance of the Xpert® Xpress Strep A for detection of *Streptococcus pyogenes* from pediatric pleural fluids

Geneviève Amaral, Vijay Gadkar, Abdulrahman Almodahka, David Goldfarb, Ryan Penny, Nicole Watson, Sukhvinder Dhaliwal, Tammy Chen, Jocelyn Srigley, Jonathan Gubbay, Miguel Imperial, and Lynne Li

Corresponding Author(s): Lynne Li, BC Children's Hospital

Review Timeline:

Submission Date:	November 14, 2025
Editorial Decision:	January 20, 2026
Revision Received:	March 20, 2026
Accepted:	March 23, 2026

Editor: Kenneth Gavina

Reviewer(s): Disclosure of reviewer identity is with reference to reviewer comments included in decision letter(s). The following individuals involved in review of your submission have agreed to reveal their identity: Tara D Fuller (Reviewer #2)

Transaction Report:

DOI: <https://doi.org/10.1128/spectrum.03703-25>

Re: Spectrum03703-25 (Performance of the Xpert® Xpress Strep A for detection of Streptococcus pyogenes from pediatric pleural fluids)

Dear Dr. Lynne Li:

Thank you for submitting your manuscript to Spectrum and for the privilege of reviewing your work. Overall, the manuscript was well written and selected for resubmission with some modifications. Below you will find the instructions from the Spectrum editorial office and feedback from the reviewers. Please ensure to fully address all reviewer comments.

Revision Guidelines

Sincerely,
Kenneth Gavina
Editor
Microbiology Spectrum

Reviewer #1 (Comments for the Author):

This is an interesting article about the use of the Cepheid Xpert Xpress Strep A qPCR test-which is FDA cleared for use on throat swabs-for the detection of Streptococcus pyogenes in pleural fluid from pediatric patients with GAS pneumonia. Cepheid's test accuracy was compared to the composite reference standard of pleural fluid culture plus an in-house LDT qPCR for S. pyogenes, and overall test performance was compared to the LDT qPCR.

Major comments:

- Methods section:

- o Retrospective and prospective samples were used, but it is generally not clear how specimens were stored prior to testing.
- o Please include the number of specimens/samples tested for each type of experiment.
- o Lines 144-147: please indicate agar plate incubation air conditions (i.e. CO₂, ambient, anaerobic).
- o Lines 150-153: it is difficult to tell how the LD-RT-PCR is performed-reagents or PCR kit used are not listed, and relevant volumes are not listed.
- o Lines 155-156: indicate the concentration of organism spiked into the pleural fluid for reproducibility and stability studies.
- o Lines 161-162: when you say the Ct was mapped to a curve of known concentrations (copies/mL), was the material used to make the curve also whole bacteria like your test material, or was the material a DNA/genomic equivalent?
- o Lines 165-169: were two different methods or criteria used to determine LOD of the LD-RT-PCR vs. the Xpert test? Hard to tell.

- Discussion section:

- o Lines 239-243: could you discuss whether it is expected that the Xpert Xpress Strep A would have any difficulty detecting any of the emm types, since this gene is not the detected target of the assay?
- o Lines 256-257: why is the *S. pyogenes* LDT PCR used on specimens that have already tested negative for *S. pyogenes* by Xpert, at your institution? (you demonstrated the LDT didn't have as good of an LOD as the Xpert...).

Minor comments:

- Abstract lines 49-50: should say "...a composite reference standard consisting of culture and a laboratory developed...". You indicate later that ALL specimens had culture AND the LDT RT-PCR, so this should be clearer here & "and/or" should not be used.
- It isn't common to use "RTPCR" to abbreviate real-time PCR. Please use "RT-PCR" or "qPCR".
- For gene names, the full gene name should be italicized. Example: *speB* and *spy1258*.
- Line 178: just checking that you did mean CFU/L and not CFU/mL?
- Line 199-201: replace "PCR" with "LD-RT-PCR" to specify which PCR you are referring to.
- Lines 211-215: uses abbreviation c/mL, but this is not used elsewhere.
- Line 250: you use "LDT-PCR" here instead of the abbreviation used throughout the manuscript.

Reviewer #2 (Comments for the Author):

Please refer to attached document

Review for: spectrum03703-25

This is an interesting article about the use of the Cepheid Xpert Xpress Strep A qPCR test—which is FDA cleared for use on throat swabs—for the detection of *Streptococcus pyogenes* in pleural fluid from pediatric patients with GAS pneumonia. Cepheid’s test accuracy was compared to the composite reference standard of pleural fluid culture plus an in-house LDT qPCR for *S. pyogenes*, and overall test performance was compared to the LDT qPCR.

Major comments:

- Methods section:
 - Retrospective and prospective samples were used, but it is generally not clear how specimens were stored prior to testing.
 - Please include the number of specimens/samples tested for each type of experiment.
 - Lines 144-147: please indicate agar plate incubation air conditions (i.e. CO₂, ambient, anaerobic).
 - Lines 150-153: it is difficult to tell how the LD-RT-PCR is performed—reagents or PCR kit used are not listed, and relevant volumes are not listed.
 - Lines 155-156: indicate the concentration of organism spiked into the pleural fluid for reproducibility and stability studies.
 - Lines 161-162: when you say the Ct was mapped to a curve of known concentrations (copies/mL), was the material used to make the curve also whole bacteria like your test material, or was the material a DNA/genomic equivalent?
 - Lines 165-169: were two different methods or criteria used to determine LOD of the LD-RT-PCR vs. the Xpert test? Hard to tell.

- Discussion section:
 - Lines 239-243: could you discuss whether it is expected that the Xpert Xpress Strep A would have any difficulty detecting any of the *emm* types, since this gene is not the detected target of the assay?
 - Lines 256-257: why is the *S. pyogenes* LDT PCR used on specimens that have already tested negative for *S. pyogenes* by Xpert, at your institution? (you demonstrated the LDT didn’t have as good of an LOD as the Xpert...).

Minor comments:

- Abstract lines 49-50: should say “...a composite reference standard consisting of culture and a laboratory developed...”. You indicate later that ALL specimens had culture AND the LDT RT-PCR, so this should be clearer here & “and/or” should not be used.
- It isn’t common to use “RTPCR” to abbreviate real-time PCR. Please use “RT-PCR” or “qPCR”.
- For gene names, the full gene name should be italicized. Example: *speB* and *spy1258*.
- Line 178: just checking that you did mean CFU/L and not CFU/mL?
- Line 199-201: replace “PCR” with “LD-RT-PCR” to specify which PCR you are referring to.
- Lines 211-215: uses abbreviation c/mL, but this is not used elsewhere.

- Line 250: you use “LDT-PCR” here instead of the abbreviation used throughout the manuscript.

The article entitled *Performance of the Xpert® Xpress Strep A for detection of Streptococcus pyogenes from pediatric pleural fluids* provides a review of the performance of Group A *Streptococcus* detection in pediatric pleural fluid using the Xpert Xpress Strep A by Cepheid GeneXpert, which has not yet been validated for invasive specimen types. The Xpert Xpress Strep A platform uses real-time PCR (RTPCR) to detect the *speB* gene, which is responsible for an endotoxin produced by Group A *Streptococcus*. Results were compared to culture alone, which demonstrates poor sensitivity, and a composite reference standard (CRS) that uses a combined method of culture and a laboratory-developed RTPCR method. This in-house RTPCR method utilizes the QuantStudio Pro instrument that targets a different Group A streptococcal gene, *spy1258*. A total of 36 clinical specimens were tested from January 2023 to April 2025, and of these, 21/36 were positive by both the composite reference standard and the Xpert Xpress. Only 8/21 positive samples grew by culture only, including the ATCC 1915 strain. The Xpert Xpress demonstrated 100% positive predictive agreement (PPA), negative predictive agreement (NPA), and overall predictive agreement (OPA) compared to the CRS.

My overall impression of the paper is that it is very well-written and demonstrates the importance of detecting Group A *Streptococcus* in invasive infections in the pediatric population. The background information and discussion are clear and concise and adequately describe the relevance of the study. The paper also notes the poor sensitivity of culture alone in detection, as well as the difference in time to detection between culture and PCR methods. Particularly important is the discussion on turn-around-time and cost efficiency as they relate to antimicrobial stewardship. The

following considerations could enhance the clarity of the paper and rigorousness of the study.

1. The Xpert demonstrated 100% reproducibility and stability, but the data for this was not shown. Numerous statistics and calculations are discussed in the body of the paper, but simplified graphs are used as figures. I would recommend a table comparing PPA, NGA, and OPA, as well as accuracy, sensitivity, and specificity as a summary statistics table.
2. It would be beneficial to briefly describe the methodology of both PCR methods in the methodology section.
3. The BioFire Film Array Pneumonia Panel detects a total of 18 bacterial, 8 viral, and 7 antimicrobial resistance targets in approximately 1 hour from bronchoalveolar lavage fluid. It may be advantageous to briefly discuss the benefits of implementing the GeneXpert Xpress into hospitals that do not have the funds or sample sizes for larger platforms that detect multiple targets in one sample with similar sensitivity, specificity, and TAT.

March 16, 2026

Dear Dr. Kenneth Gavina,

Re: Spectrum 03703-25: Performance of the Xpert® Xpress Strep A for detection of *Streptococcus pyogenes* from pediatric pleural fluids

Thank you for your letter dated January 20, 2026 regarding our submission to Microbiology Spectrum.

My colleagues and I greatly appreciate the detailed review of our manuscript and the constructive feedback provided by yourself and the reviewers. As requested, we have submitted a revised manuscript, both a version with in-track and highlighted changes, and a final, clean version of the manuscript. We would also like to respond to each reviewer's comments on a point-by-point basis as indicated in blue text below. Line numbers correspond to the final, clean manuscript. We hope that our responses and revised manuscript will resolve outstanding concerns for our study.

We confirm that this work is original and not published elsewhere, nor is it currently under consideration at another journal. We warrant that all named authors have seen and approved the revised manuscript and contributed significantly to this work.

Thank you for your consideration of our revised manuscript. We greatly look forward to your response.

Sincerely,

Lynne Li, MD, FRCPC

Medical Microbiologist & Infection Control Physician, BC Children's and Women's Hospital & Health Centre, Vancouver, British Columbia, Canada

Email: lynne.li@cw.bc.ca

On behalf of co-authors: Geneviève Amaral, Vijay J. Gadkar, Abdulrahman Almodahka, David M. Goldfarb, Ryan Penny, Nicole Watson, Suk Dhaliwal, Tammy Chen, Jocelyn A. Srigley, Jonathan B. Gubbay, Miguel Imperial

Reviewer Response:

Reviewer #1 (Comments for the Author):

This is an interesting article about the use of the Cepheid Xpert Xpress Strep A qPCR test-which is FDA cleared for use on throat swabs-for the detection of Streptococcus pyogenes in pleural fluid from pediatric patients with GAS pneumonia. Cepheid's test accuracy was compared to the composite reference standard of pleural fluid culture plus an in-house LDT qPCR for *S. pyogenes*, and overall test performance was compared to the LDT qPCR.

Major comments:

- Methods section:

- Retrospective and prospective samples were used, but it is generally not clear how specimens were stored prior to testing.

Thank-you for indicating the need for clarification. Both prospective (n=7) and retrospective (n=29) samples were stored at -80°C prior to testing.

Line 133: We have added the statement that samples were stored at -80°C prior to testing.

- Please include the number of specimens/samples tested for each type of experiment.

Thank-you for highlighting the need for clarification. We have added the following:

- 2.2 Test performance: Line 137: n=36 samples tested
- 2.3 Reproducibility and stability: Line 165: Two strains listed; Line 168: storage conditions clarified
- 2.4 Analytical sensitivity: Line 180 & 185: Six dilutions were performed. Line 181-184: The concentration of each dilution beginning from highest concentration was: 103.34 c/mL, 51.61 c/mL, 25.83 c/mL, 12.9 c/mL, 12.9 c/mL, 6.45 c/mL and 3.22 c/mL. Five replicates were performed for each dilution.
- 2.5 Analytical specificity and reactivity:
 - Line 191-194: The text specifies n=10 non-*S. pyogenes* species were used to test analytical specificity. We have further specified that two clinical samples were used (one of which was polymicrobial accounting for three organisms), and the remaining six samples were spiked isolates (single organism each).
 - Line 201: For reactivity, 6 different emm types of *S. pyogenes* were used (indicated in text)

- Lines 144-147: please indicate agar plate incubation air conditions (i.e. CO₂, ambient, anaerobic).

Thank-you for highlighting the need for clarification. Our institutional protocol uses the following incubation conditions for blood agar (BA), chocolate agar (CHOC), MacConkey agar (MAC), and Brucella anaerobic agar (BAN):

Processing: Refer to Flowchart #1

BA	CO ₂	35°C/ 48 h
CHOC	CO ₂	35°C/ 48 h
MAC	O ₂	35°C/ 24 h
BAN	AnO ₂	35°C/ 5 days
Peds Plus bottle and/or BHI (see flowchart #1)		

Line 154-157: Agar plate incubation air conditions have been added.

- Lines 150-153: it is difficult to tell how the LD-RT-PCR is performed-reagents or PCR kit used are not listed, and relevant volumes are not listed.

Thank-you for the suggestion. The LD-qPCR targeting *spy1258* gene is performed using the following reagents: 5 µL of extracted patient sample, 20 µL Master Mix [comprised of : TaqMan™ Fast Advanced Master Mix; Primer/Probe mix using previously published CDC-GAS primers/probe: <https://www.cdc.gov/strep-lab/media/pdfs/2024/06/oligonucleotides-triplex-rtPCR.pdf>]

The LD-qPCR was performed under the following conditions:

Temperature Cycles:

95°C -2 min

95°C -3 sec

60°C -20 sec

} repeat 45 x

Lines 148-152: We have added additional information outlining the above PCR conditions.

- Lines 155-156: indicate the concentration of organism spiked into the pleural fluid for reproducibility and stability studies.

Thank-you for highlighting the need for clarification. The *S. pyogenes* ATCC 1915 strain was spiked in the negative pleural fluid at concentration of 1:10 from the stock 0.5 McFarland.

Lines 165-166: Updated to reflect this addition.

- Lines 161-162: when you say the Ct was mapped to a curve of known concentrations (copies/mL), was the material used to make the curve also whole bacteria like your test material, or was the material a DNA/genomic equivalent?

Thank-you for highlighting the need for clarification. The calibration curve to determine the concentration (copies/mL) of the gDNA extracted from the whole bacterium was generated using a double stranded, synthetic DNA fragment (gBlock™, IDT, Coralville, IA) containing the *spy1258* target. This calibration curve (Ct vs copies/mL) provided the actual copies of the target (*spy1258*) in the gDNA extracted from the whole bacterium. In other words, a whole bacterium was used and not genomic equivalents.

Lines 175-179: The text has been updated to reflect these details.

- Lines 165-169: were two different methods or criteria used to determine LOD of the LD-RT-PCR vs. the Xpert test? Hard to tell

Thank-you for highlighting the need for clarification. For Xpert, we performed a 95% hit rate analysis because we were confirming detection of a commercial test at a predefined concentration using replicate testing of the cartridges. For the LD-qPCR, we applied probit analysis to statistically model the concentration–response relationship based on the dilution sequence in order to estimate the 95% detection limit with confidence intervals.

Lines 180-189: The text has been revised to clarify that GeneXpert LoD was determined by 95% hit rate analysis, while LD-qPCR was determined by probit analysis.

- Discussion section:

- Lines 239-243: could you discuss whether it is expected that the Xpert Xpress Strep A would have any difficulty detecting any of the emm types, since this gene is not the detected target of the assay?

Thank-you for raising this interesting point. The *speB* gene is highly conserved among *S. pyogenes* and thus it would not be expected that the Xpert® Xpress Strep A would have difficulty detecting any *emm* type. For the purposes of our validation and to ensure comprehensive analysis, we felt it was appropriate to include several *emm* types. There are rare reports of *speB*-deficient mutants (identified from an *emm82* and *emm11* type) documented in the literature (see updated references 28 and 29 in text). Thus, we felt it was worthwhile to include this data to highlight the possibility of a false-negative result, and to add to the literature for further epidemiological understanding. As such we have made the following modification:

Lines 264-269: We have added a comment referencing prior studies with *speB* deficient mutants.

Lines 281-286: We have clarified that our laboratory uses the LD-qPCR targeting the *spy1258* gene, if pleural fluids test negative on the Xpert® assay, which would help enable detection of the rare *speB* mutants.

- Lines 256-257: why is the *S. pyogenes* LDT PCR used on specimens that have already tested negative for *S. pyogenes* by Xpert, at your institution? (you demonstrated the LDT didn't have as good of an LOD as the Xpert...).

Thank-you for highlighting this point of clarification. At the time of this study, we were prospectively evaluating both testing methods, and thus had not accumulated the data to demonstrate which method had superior analytical sensitivity. As such, to ensure a rigorous evaluation of the Xpert®, we performed both methods on all samples.

Minor comments:

- Abstract lines 49-50: should say "...a composite reference standard consisting of culture and a

laboratory developed...". You indicate later that ALL specimens had culture AND the LDT RT-PCR, so this should be clearer here & "and/or" should not be used.

Thank-you for highlighting the wording inconsistency. We have made the following modifications:

Line 50: has been updated to read as: "...culture and a laboratory developed..."

Line 544-545: Table 1 title has been updated in keeping with this "...*spy1258* and culture..."

- It isn't common to use "RTPCR" to abbreviate real-time PCR. Please use "RT-PCR" or "qPCR".

Thank-you for highlighting this accepted abbreviation. The text has been updated with RTPCR changed to qPCR (highlighted) throughout.

- For gene names, the full gene name should be italicized. Example: *speB* and *spy1258*.

Thank-you for correcting the accepted nomenclature. The text has been updated with fully italicized gene names for *speB*, and *spy1258* (highlighted) throughout.

- Line 178: just checking that you did mean CFU/L and not CFU/mL?

Thank-you for identifying this typo. We have made the following modification:

Line 199: The text has been corrected to CFU/mL

- Line 199-201: replace "PCR" with "LD-RT-PCR" to specify which PCR you are referring to.

Thank-you for highlighting the need for clarification. The text has been updated correspondingly:

Lines 221-223 have been updated to LD-qPCR, in keeping with the above modification.

- Lines 211-215: uses abbreviation c/mL, but this is not used elsewhere.

Thank-you for identifying this typo. We have updated the text to clarify this abbreviation as copies/mL (c/mL).

Line 173: We have specified copies/mL (c/mL).

- Line 250: you use "LDT-PCR" here instead of the abbreviation used throughout the manuscript.

Thank-you for identifying this typo. The abbreviation has been corrected to LD-RT-PCR.

Line 275: The text has been updated with LDT-PCR changed to LD-qPCR.

Reviewer 2:

1. The Xpert demonstrated 100% reproducibility and stability, but the data for this was not shown. Numerous statistics and calculations are discussed in the body of the paper, but simplified graphs are used as figures. I would recommend a table comparing PPA, NGA, and OPA, as well as accuracy, sensitivity, and specificity as a summary statistics table.

Thank-you for the suggestion. In Line 227, we have included a statement directing the reader to Table 3 for summary statistics.

We have added Table 3 (see below), which summarizes the PPA, NPA, OPA, accuracy, sensitivity, and specificity statistics for the Xpert® as compared to the composite reference standard (CRS). We have included the PPA, NPA, and OPA for Xpert® vs. Culture alone, however given that we expect the

molecular test to be more sensitive than culture, we have not included accuracy, sensitivity, or specificity for this comparison to avoid biasing the results. The kappa statistic has been included to support interpretation of OPA.

Table 3. Summary of Xpert® Xpress Strep A performance statistics and test characteristics as compared to composite reference standard (CRS; LD-RT-PCR and culture) vs. culture alone

	PPA (%)	NPA (%)	OPA (%) [κ]	Accuracy (%)	Sn (%)	Sp (%)
Xpert® vs. CRS	100	100	100 [1]	100	100	100
Xpert® vs. culture	100	54	64 [0.3]	N/A	N/A	N/A

Abbreviations: PPA=positive percent agreement; NPA=negative percent agreement; OPA=overall percent agreement; κ =kappa statistic; Sn=sensitivity; Sp=specificity; N/A=not applicable

2. It would be beneficial to briefly describe the methodology of both PCR methods in the methodology section.

Thank-you for the suggestion. We have added further descriptions regarding LD-qPCR and the Xpert method as indicated above to Reviewer #1's comments.

Lines 137- 140: Additional explanation of the GeneXpert system has been included in the text.

Lines 148-152: We have added additional information outlining the above PCR conditions.

3. The BioFire Film Array Pneumonia Panel detects a total of 18 bacterial, 8 viral, and 7 antimicrobial resistance targets in approximately 1 hour from bronchoalveolar lavage fluid. It may be advantageous to briefly discuss the benefits of implementing the GeneXpert Xpress into hospitals that do not have the funds or sample sizes for larger platforms that detect multiple targets in one sample with similar sensitivity, specificity, and TAT.

Thank-you for the suggestion to compare with other commercial multiplex panels that have the same targets. In our jurisdiction, the Biofire FilmArray Pneumonia Panel is not licensed (has not been approved by Health Canada). The Biofire Spotfire Respiratory/Sore Throat Panel is recently licensed in our jurisdiction and includes a GAS target, however, it has a higher upfront/testing cost compared to Xpert, with several less relevant targets included for pediatric empyema/parapneumonic effusion.

Line 295 – 306: We have added a discussion regarding cost feasibility of GAS vs. commercial multiplex assays, particularly in smaller centers, as well as the licensing considerations specific to our location.

Re: Spectrum03703-25R1 (Performance of the Xpert® Xpress Strep A for detection of Streptococcus pyogenes from pediatric pleural fluids)

Dear Dr. Lynne Li:

I am delighted to share that your manuscript has been accepted, and I am forwarding it to the ASM production staff for publication. Your paper will first be checked to make sure all elements meet the technical requirements. ASM staff will contact you if anything needs to be revised before copyediting and production can begin. Otherwise, you will be notified when your proofs are ready to be viewed.

Sincerely,
Kenneth Gavina
Editor
Microbiology Spectrum